# Dry Climate Filters Gymnosperms but Not Angiosperms through Seed Mass

Yang Qi [1], Hongyan Liu [1],*, Chongyang Xu [2], Jingyu Dai [1] and Biao Han [3]

[1] College of Urban and Environmental Sciences and MOE Laboratory for Earth Surface Processes, Peking University, Beijing 100871, China; qi_yang@pku.edu.cn (Y.Q.); daijingyu@pku.edu.cn (J.D.)

[2] Faculty of Agriculture, Food and Environment, The Hebrew University of Jerusalem, Rehovot 7610001, Israel; chongyang.xu@mail.huji.ac.il

[3] Key Laboratory of State Forestry and Grassland Administration Conservation and Utilization of Warm Temperate Zone Forest and Grass Germplasm Resources, Shandong Provincial Center of Forest and Grass Germplasm Resources, Jinan 250014, China; hanbiaook831228@163.com

* Correspondence: lhy@urban.pku.edu.cn

**Abstract:** In the context of climate change in recent years, the fate of woody plant seed has an important impact on forest regeneration. Seed mass is an important reproductive strategy of plants. There are huge differences between gymnosperms (mainly conifers) and angiosperms (flowering plants) in terms of reproduction and hydraulic strategies; however, little is known about changes in seed mass along climate aridity gradients between taxonomical groups such as gymnosperms and angiosperms, which limit our understanding on the fate of woody plants under warming-induced climate drying. We collected seed mass data from a total of 2575 woody plant individuals, including 145 species of gymnosperms and 1487 species of angiosperms, across different climatic zones in China. We mapped the distribution pattern of gymnosperm and angiosperm seed mass in China, with angiosperms being maximal near the 400 mm iso-precipitation line. Our phylogenetic analysis results show that seed mass exhibited significant phylogenic signals ($p < 0.001$) and was also strongly influenced by functional traits (growth type, fruit type, and dispersal mode). The results of linear regression and hierarchical partitioning analysis showed a stronger correlation between gymnosperm seed mass and environmental factors, and a higher independent aridity index effect on gymnosperm seed mass than angiosperm seed mass. The different patterns of seed mass along a climate aridity gradient between gymnosperms and angiosperms may point to different future fates for these two taxonomic groups, while the higher sensitivity of gymnosperm seed mass to environmental conditions may reduce their reproductive rate under the background of climate warming and drying.

**Keywords:** seed mass; gymnosperms; angiosperms; climatic dryness; aridity gradient

## 1. Introduction

Seed plants have incomparable competitive advantages in breeding, life history, and adaptation to harsh environments [1]. As an indispensable stage in the life history of seed plants, seeds have been an important subject that have received extensive attention since the last century [2]. Seed mass, defined as the dry weight of seeds excluding dispersal structures but including the testa, is the most important, widely used, and easily measured seed trait [3–5]. Thus, it is indisputable that seed mass changes and that seed mass variety is an adaptation of plants driven by genetic selection in response to the natural environment [6]. Globally, the mass of seeds differs by more than ten orders of magnitude [6,7]. Meanwhile, cross-species studies show that growth form is the strongest correlate of seed size [6]. Many evidences in modern and ancient fossils indicate that gymnosperms and angiosperms have significant differences in seed mass [6,8]. Studies on sample plots, local scales, and global scales have shown that seed mass is also affected by changing temperature and

precipitation [9–11]. Seed development and maturation, which are temperature- and moisture-dependent processes, affect seed mass [12–14].

In the past century, drought has been the basis for many large-scale forest deaths [15,16], and rising global temperatures have exacerbated the changes in forests caused by drought. In the context of climate change in recent years, the fate of woody plant seed has an important influence on forest regeneration [17–19]. Seed mass is an important reproductive strategy of plants, but it is still unknown whether seed mass, together with other strategies, determines the survival strategy of plants in response to climate drying. There are huge differences between gymnosperms (mainly conifers) and angiosperms (flowering plants) in terms of reproduction [20] and hydraulic strategies [15,21]. Darwin referred to angiosperm radiation as an "abhorrent mystery" [22]. This phenomenon has been explained as the higher trait flexibility of angiosperms in breeding systems, pollen, and geography than gymnosperms [23,24]. Moreover, gymnosperms have more drought-tolerant hydraulic properties than angiosperms [21]. Under water stress, gymnosperms have a greater hydraulic safety margin and are less prone to stem embolisms [15,25]. Some gymnosperms can promote the closure of stomata and reduce water loss through high levels of the hormone abscisic acid (ABA) under water stress [25]. However, there is currently a lack of research on the differences in seed mass responses to climatic water stress between gymnosperms and angiosperms. Therefore, we hypothesized that gymnosperms and angiosperms respond differently to climate factors, which is also reflected in changes in seed mass.

To test the hypothesis, China was chosen as our study region, because the climate in China is diverse, so it is possible to sample a wide range of climate aridity regimes, including arid, semiarid, semi-humid, and humid regions. We collected data on a total of 2575 individuals from various sites, including 145 species of gymnosperms and 1487 species of angiosperms.

## 2. Materials and Methods

### 2.1. Sources of Seed Mass and Species Distribution Data

Woody plant seed mass was obtained from the book "Chinese Woody Plant Seeds" [26] and the Shandong Forest and the Grass Germplasm Resource Center. The seed mass data in the book were taken from forestry survey data from various places. The seed masses were reported in the article as the thousand-seed weight of dry seeds, so our statistical analysis was also based on the thousand-seed weight. Species names were checked against The Plant List (www.theplantlist.org, accessed on 1 March 2020). Subspecies and varieties were not recognized in the analyses. The growth forms of species (trees, shrubs, shrubs or small trees, and lianas) and fruit type (fleshy fruit, dry fruit, aggregate fruits, collective fruits, and cone) were collected and supplemented in the Flora of China (http://www.efloras.org/flora_page.aspx?flora_id=2, accessed on 1 March 2020). Seed dispersal methods (gravity dispersal, wind dispersal, and animal dispersal) were collected from published literature searches.

The distribution data of woody plants in China were taken from the latitude and longitude data of plant sample collection data points in the online sharing platform of the National Plant Specimen Resource Center (https://www.cvh.ac.cn/, accessed on 1 March 2020). The location where the species sample was collected was considered as the distribution location of the species.

Since this article mainly focuses on the distribution of inter-species seed mass, the seed mass of the same species was averaged first, and the data values with large intra-species differences were eliminated according to the variance. Species distribution data and seed mass data were matched, and the species with both distribution data and seed quality data were retained. The distribution data were transformed into raster data of 50 km × 50 km, and the average value of the seed mass of all species existing in each pixel was calculated as the seed mass value of the pixel, and no weight was assigned to the species. At the same time, the maximum and minimum seed masses in each pixel were extracted for subsequent analysis.

Data were collected for a total of 2575 individuals, of which 179 were gymnosperms and 2396 were angiosperms, containing seed mass values for different distribution locations of the same species. The data covered 145 species of gymnosperms and 1487 species of angiosperms, covering more than 2/3 of all the woody plant families in China [27].

## 2.2. Climatic Metrics

It is generally believed that the spatial variation in seed mass is affected by temperature and precipitation. In order to compare with previous studies, this study chose the mean annual temperature (MAT) to characterize the ambient temperature, and selected the annual precipitation (MAP) and precipitation of the warmest quarter (MPWQ). The aridity index (AI) was used to characterize regional water availability. Since seeds can be spread by wind, the wind speed in this area was selected as the influencing factor of seed mass. At the same time, considering that light also has an impact on seed production, the follow-up analysis also included solar radiation (SRAD).

The modern mean annual temperature (MAT), wind speed, precipitation of the warmest quarter (MPWQ), annual precipitation (AP), and solar radiation (SRAD) data of each sampling site were obtained from WorldClim Global Climate Data version 2 (www.worldclim.org, accessed on 1 March 2020) [28]. The spatial resolutions of data were 10 min. The aridity index was calculated as the ratio of mean annual precipitation and potential evapotranspiration for each site; hence, the smaller magnitude in this value indicated the drier environments and vice versa. The potential evapotranspiration (PET) data (version 3) were obtained from the Consultative Group on International Agricultural Research—Consortium for Spatial Information (CGIAR-CSI, www.cgiarcsi.community, accessed on 1 March 2020) [29]. All environmental factors first extracted the research area from the global raster data and then resampled it to a data resolution of 50 km $\times$ 50 km.

## 2.3. Methods

Due to the large difference in the overall seed mass, in a subsequent analysis, a natural logarithmic transformation was applied to the seed mass data. To test for the phylogenetic clustering of seed mass, we generated a phylogenetic tree using the 'V.PhyloMaker' package in R 4.1.2 [30] (Figure S1), the backbone of which is a phylogeny of 79,881 taxa of seed plants, developed by Smith and Brown [31]. The constructed phylogenetic tree contained 1580 species in our database. The phylogenetic signal in the seed mass was tested by estimating Blomberg's K [32] and Pagel's λ [33] with the 'phylosig' function in the R package 'phytools' [34]. In IBM SPSS 20.0, the Kolmogorov–Smirnov test was used to test whether the gymnosperm and angiosperms seed masses conform to the normal distribution. In addition, the Mann–Whitney U test was used to compare the seed mass difference between gymnosperms and angiosperms.

The spatial analysis and mapping of seed mass geographic patterns were implemented in ArcGIS 10.2. The effect of different functional traits on seed mass was studied by one-way analysis of variance (ANOVA) in SPSS 20.0. Ordinary linear regression, operated by the 'lm' package, was used to analyze the relationship between seed mass and environmental factors, and the significance levels of all the regression relationships were tested with the *t*-test. Additionally, hierarchical partitioning analysis with the R package 'hier.part' was used to quantify the strength of the environment's influence on the spatial variability in seed mass, compare the relative contributions between factors, and determine the main driving climate factor. Its use in ecology and conservation is increasing due to its complementary role to multiple regression analysis [35].

## 3. Results

### 3.1. Geographic Patterns of Seed Mass

The average seed mass of gymnosperms in the grid decreased from the south to the north and northwest of China (Figure 1a), which corresponded to the decreasing gradient of temperature and precipitation, from the humid tropical and subtropical climate in the

south to the cold desert climate in the northwest. Gymnosperm seed mass was the largest in Hainan Island and southern Yunnan, China, but relatively small in northeastern China and Inner Mongolia. Gymnosperm seed mass-changed significantly near the 400 mm isoprecipitation line (Figure 1a).

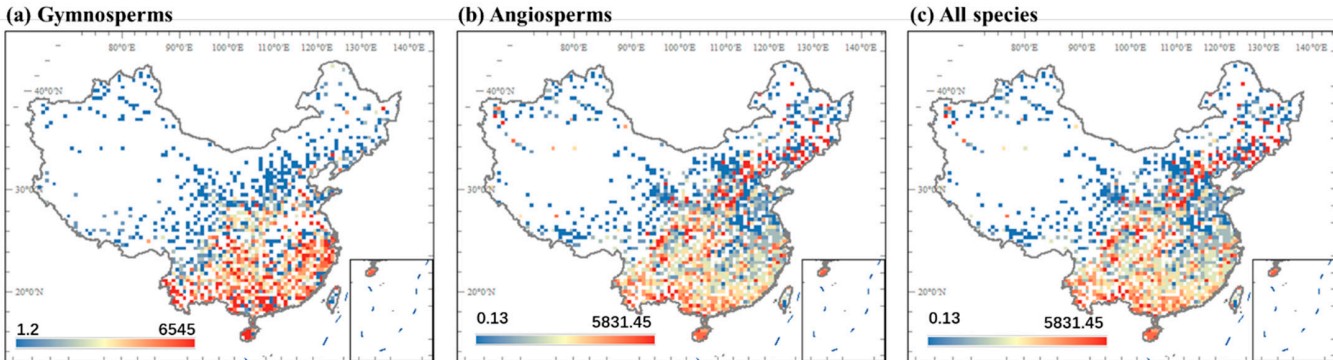

**Figure 1.** The geographic patterns in the seed mass of woody species in China. (**a**) Gymnosperms; (**b**) angiosperms; (**c**) all species. The value increased from blue to red. The spatial resolution was 50 × 50 km.

The geographical patterns of the average seed masses of angiosperms and all species in the grid were different from those of gymnosperms. Although the average seed mass in southern China was higher than that in northern China, the seed mass of angiosperms was the largest near the 400 mm isoprecipitation line (Figure 1b,c).

### 3.2. The Influence of Evolutionary History on Seed Mass

The weight of 1000 seeds of the woody plants ranged from 0.01 g (*Deutzia parviflora*) to 78,000 g (*Aesculus wangii*), denoting a difference of six orders of magnitude. Blomberg's K and Pagel's λ calculations based on the dated phylogenetic tree showed that seed mass exhibited a significant phylogenic signal (*p* < 0.001, Table 1).

**Table 1.** Phylogenetic signals (Blomberg's *K* and Pagel's λ) of seed mass.

| | | Blomberg's K | | Pagel's λ | |
|---|---|---|---|---|---|
| **Seed Mass** | **n** | **K** | **P** | **λ** | **P** |
| | 1580 | 0.150 | 0.001 | 0.777 | 0.000 |

The results of the Mann–Whitney U test showed that the mass of gymnosperm and angiosperm seeds did not accord with normal distribution rates. Angiosperms have a larger distribution of seed mass than gymnosperms (Figure 2). Gymnosperm seed mass was positively skewed, and the number of seeds was higher at the site of small seed mass (Table 2). The distribution of angiosperm seed mass was negatively skewed, and the data volume of large seed mass was higher (Table 2). The kurtosis of gymnosperm and angiosperm seed mass distribution was relatively steep (Table 2). The results of the Mann–Whitney U test show that there was a significant difference between the seed mass of gymnosperms and angiosperms in the overall comparison (*p* < 0.05, Figure 2).

**Table 2.** Skewness and kurtosis of gymnosperm and angiosperm seed mass.

| | Skewness | | Kurtosis | |
|---|---|---|---|---|
| | **Value** | **Standard Error** | **Value** | **Standard Error** |
| Gymnosperms | 0.919 | 0.182 | 0.474 | 0.361 |
| Angiosperms | −0.152 | 0.048 | 0.441 | 0.096 |

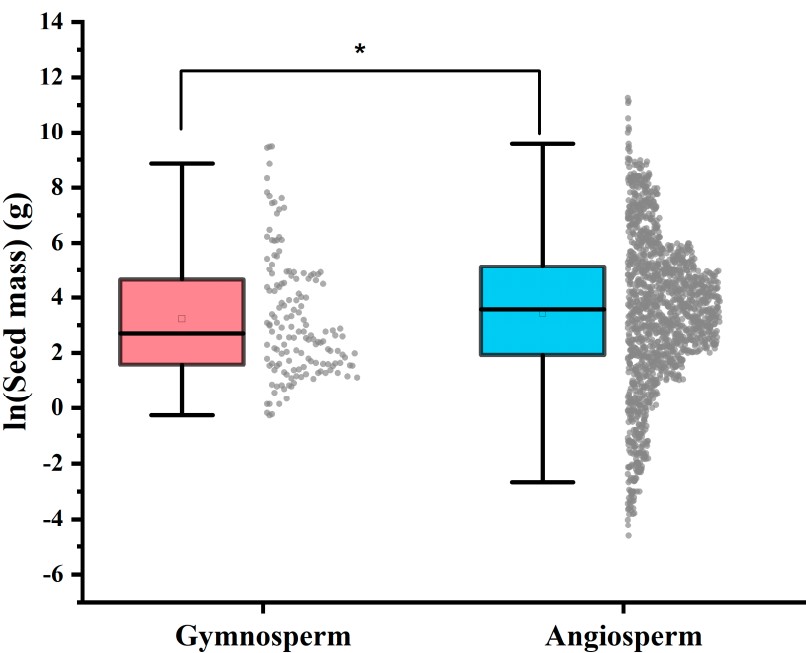

**Figure 2.** Box diagrams of seed mass distribution. There was a significant difference between gymnosperms ($n$ = 175) and angiosperms ($n$ = 2572) in the overall comparison ($p < 0.05$). Red indicates gymnosperms, blue indicates angiosperms. The horizontal lines represent the 25%, 50%, and 75% quantiles. The vertical lines show the 5% and 95% quantiles. Asterisks above the horizontal line indicate significant differences among gymnosperm and angiosperm ($p < 0.05$).

### 3.3. The Effects of Functional Traits on Seed Mass

The seed mass of the trees was significantly larger than that of small trees or shrubs ($p < 0.05$), and significantly larger than that of shrubs ($p < 0.05$, Figure 3A). There was no significant difference in the seed mass of shrubs and lianas ($p > 0.05$, Figure 3A). The seed mass of the fleshy fruit was significantly greater than that of the aggregate fruit ($p < 0.05$, Figure 3B) and significantly greater than that of the dry fruit ($p < 0.05$, Figure 3B), and the seed mass of the collective fruit was significantly lower than that of other types of fruit ($p < 0.05$, Figure 3B). The seed mass of the cones was between those of fleshy and collective fruits. Gravity dispersed seeds had the highest mass, followed by animals, and wind-borne seeds had the least mass ($p < 0.05$, Figure 3C).

### 3.4. Relationships between Climatic Indices and Woody Plant Seed Mass

Gymnosperm and angiosperm seed masses were linearly correlated with environmental factors (Figure 4a–f, Table S1). The results showed that the seed masses of gymnosperms and angiosperms had a highly significant negative correlation with wind speed and solar radiation ($p < 0.01$, Figure 4b,c, Table S1), and a highly positive significant correlation with other environmental factors, including mean annual temperature, precipitation of the warmest quarter, annual precipitation, and the aridity index ($p < 0.01$, Figure 4a,d–f, Table S1). Gymnosperm seed mass had the strongest correlation with annual precipitation (maximum $R^2$, Figure 4e, Table S1) and the weakest correlation with solar radiation (minimum $R^2$, Figure 4b, Table S1). Angiosperm seed mass had the strongest correlation with precipitation of the warmest quarter (maximum $R^2$, Figure 4d, Table S1) and the weakest correlation with solar radiation (minimum $R^2$, Figure 4b, Table S1). In addition to solar radiation, annual average temperature, wind speed, precipitation of the warmest quarter, annual precipitation, and the aridity index were all more correlated with gymnosperms than with angiosperms ($R^2$ was higher, Figure 4a–f, Table S1).

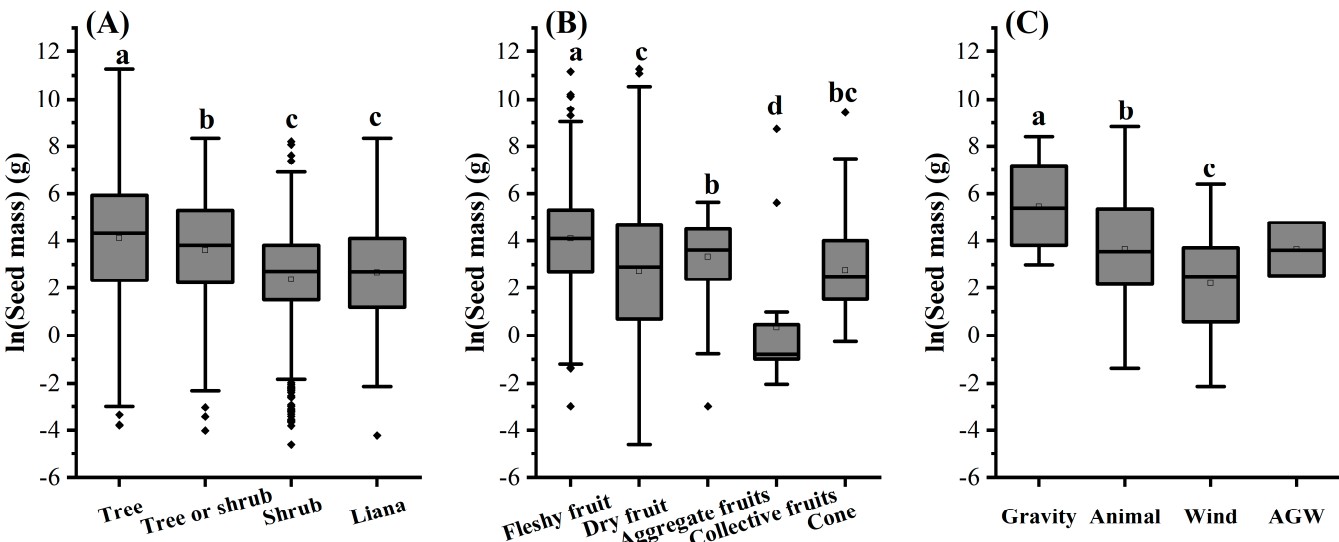

**Figure 3.** Relationship between seed mass and functional traits in woody plants. Differences in seed mass of species with (**A**) different growth forms (tree, shrub or small tree, shrub, and liana); (**B**) different fruit types (fleshy fruit, dry fruit, aggregate fruits, collective fruits, and cone); (**C**) different seed dispersal modes (gravity dispersal, animal dispersal, wind dispersal, and a variety of seed dispersal methods). The horizontal lines represent the 25%, 50%, and 75% quantiles. The vertical lines show the 5% and 95% quantiles. Diamond-shaped points represent outliers. The different lowercase letters indicate significant differences in the seed mass ($p < 0.05$).

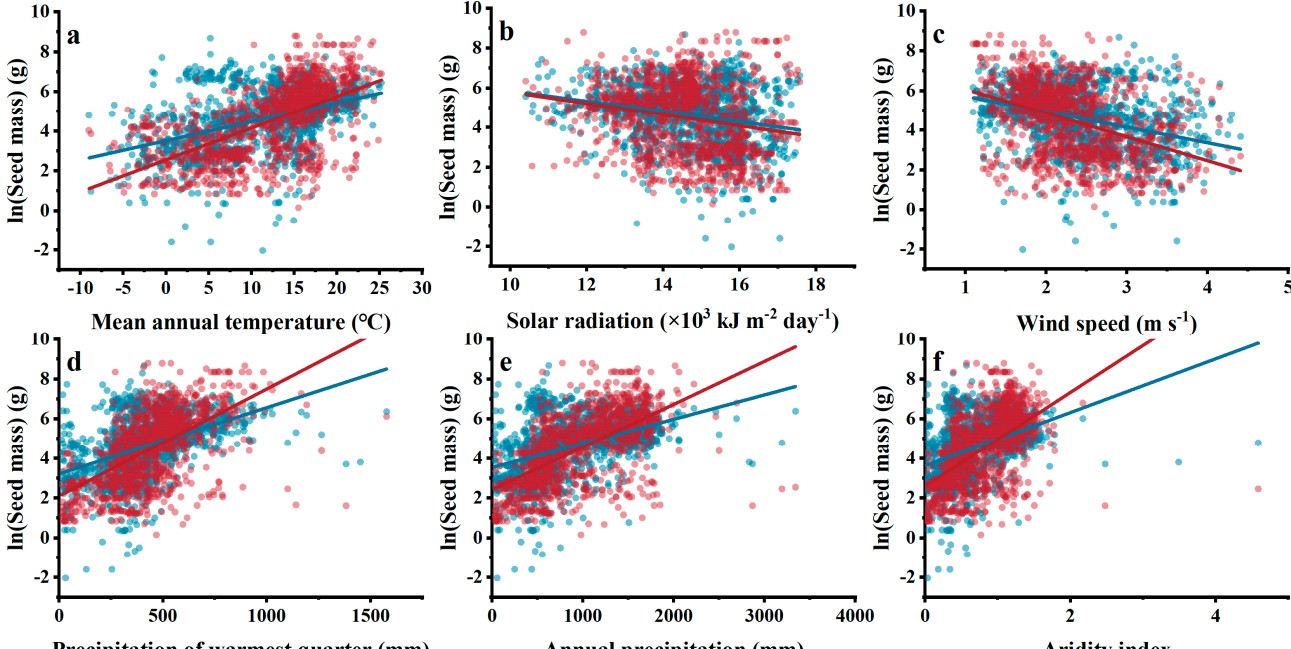

**Figure 4.** Relationships between the seed mass of woody plants and (**a**) Mean annual temperature; (**b**) Solar radiation; (**c**) Wind speed; (**d**) Precipitation of the warmest quarter; (**e**) Annual precipitation and (**f**) Aridity index. The red dots and fitted straight line indicate the gymnosperms, and the blue dots and fitted straight lines indicate the angiosperms. The solid lines indicate significant linear regression ($p < 0.01$).

The upper limit ($G_{max}$) and lower limit ($G_{min}$) of the gymnosperm seed mass data range increased significantly with an increase in the MAT, MPWQ, and MAP ($p < 0.05$, Figure 5a,d–e, Table S1). The upper limit ($G_{max}$) of the seed mass data range decreased

significantly with an increase in the SRAD and wind speed ($p < 0.05$, Figure 5b,c, Table S1). The upper limit ($A_{max}$) of the angiosperm seed mass data range increased significantly with an increase in the MAT, MPWQ, and MAP ($p < 0.05$, Figure 6a,d–e, Table S1), and decreased significantly with an increase in the SRAD and wind speed ($p < 0.05$, Figure 6b,c Table S1). The lower limit ($A_{min}$) of the angiosperm seed mass data range decreased significantly with an increase in the MAT, MPWQ, and MAP ($p < 0.05$, Figure 6a,d,e, Table S1), and increased significantly with an increase in the SRAD and wind speed ($p < 0.05$, Figure 6b,c, Table S1).

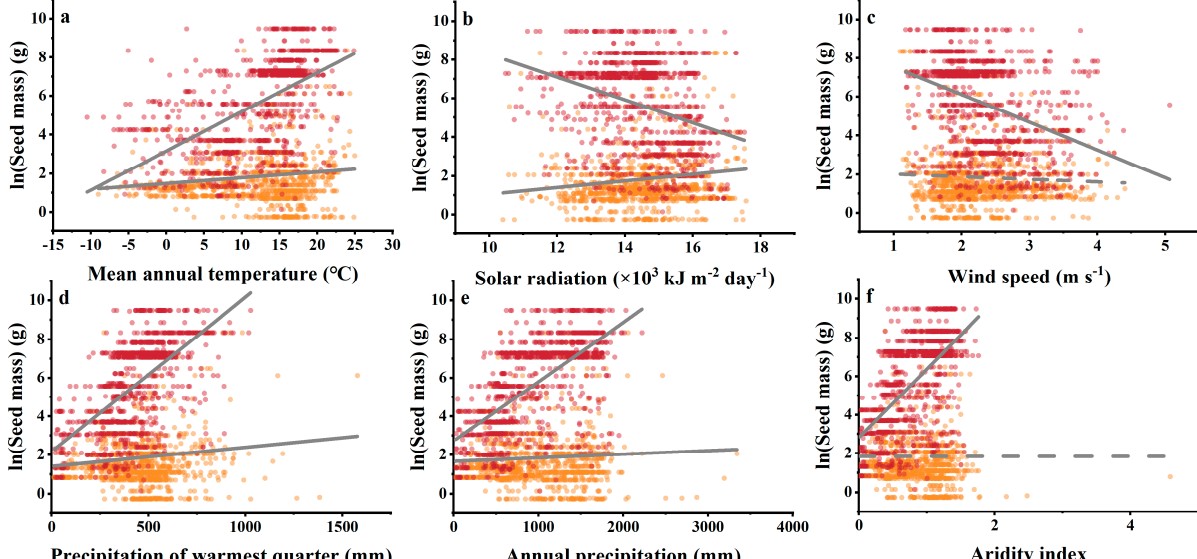

**Figure 5.** Relationships between the minimum and maximum seed masses of gymnosperms and (**a**) Mean annual temperature; (**b**) Solar radiation; (**c**) Wind speed; (**d**) Precipitation of the warmest quarter; (**e**) Annual precipitation and (**f**) Aridity index. The red dots indicate the maximum value of gymnosperm seed mass and the orange dots indicate the minimum value. The solid line indicates a significant linear regression ($p < 0.05$), while the dashed line indicates an insignificant linear regression ($p > 0.05$).

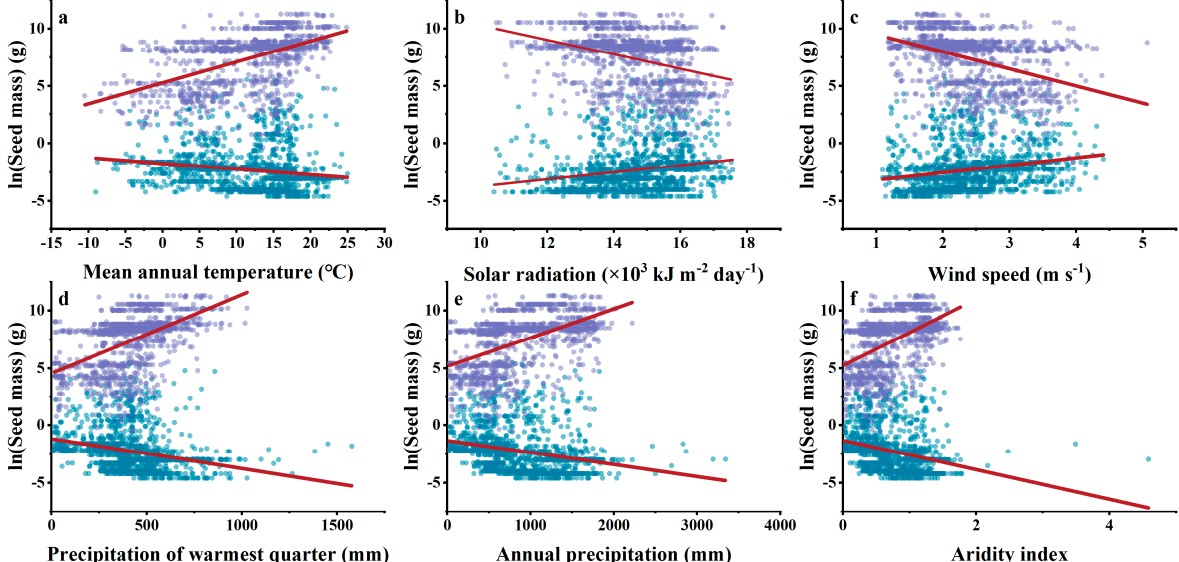

**Figure 6.** Relationships between the minimum and maximum seed masses of angiosperms and (**a**) Mean annual temperature; (**b**) Solar radiation; (**c**) Wind speed; (**d**) Precipitation of the warmest quarter; (**e**) Annual precipitation and (**f**) Aridity index. The purple dots indicate the maximum value of the angiosperm seed mass and the blue dots indicate the minimum value of the angiosperm seed mass. The solid line indicates a significant linear regression ($p < 0.05$).

As the MAT increased, the data range of the gymnosperm seed mass gradually increased (the slope of $G_{max}$ was greater than that of $G_{min}$) (Figure 5a, Table S1), and the same trend was shown as the MPWQ and MAP increased (Figure 5d,e, Table S1). The data range of the angiosperm seed mass tended to increase with increases in the MAT, MPWQ, MAP, and AI (the slope of $A_{max}$ was greater than that of $A_{min}$) (Figure 6a,d–f, Table S1). As the wind speed increased, the data range gradually decreased (the slope of $A_{max}$ was smaller than that of $A_{min}$) (Figure 6c, Table S1). With the increase in the SRAD, the distribution range of gymnosperm and angiosperm seed mass decreased (Figures 5b and 6b, Table S1).

*3.5. Contribution of Individual Environmental Factors to Seed Mass*

The results of hierarchical partitioning showed that MAP (25.1%) had the highest independent effect on the gymnosperm seed mass, followed by MAT (22.7%) and MPWQ (21.1%) (Figure 7a). The MPWQ (24.2%) had the highest independent effect for the angiosperm seed mass, followed by the MAT (21.7%) (Figure 7b). The SRAD had the lowest independent effect for gymnosperm (2.1%) and angiosperm (6.9%) seed masses (Figure 7).

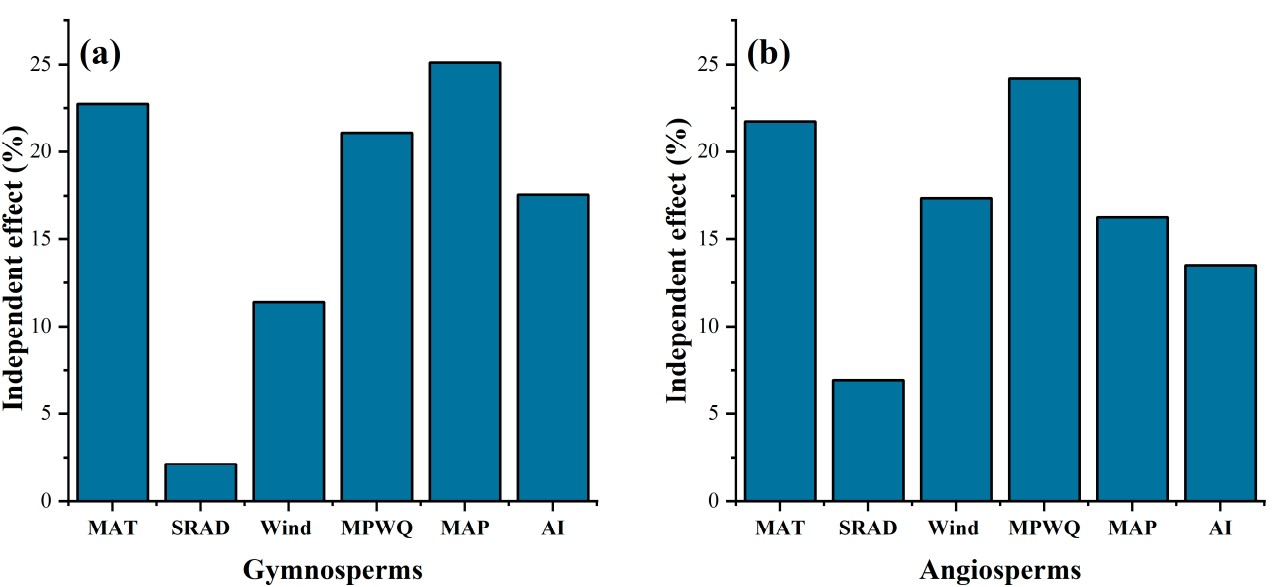

**Figure 7.** Independent effects of environmental factors on seed mass. (**a**) Gymnosperms. (**b**) Angiosperms. MAT: mean annual temperature. Wind: wind speed. MPWQ: precipitation of the warmest quarter. SRAD: solar radiation. MAP: annual precipitation. AI: aridity index.

## 4. Discussion

*4.1. Distribution and Influencing Factors of Seed Mass*

The seed mass of Chinese woody plants (including gymnosperms and angiosperms) tends to decrease from the southeast to the northwest of China. Different from gymnosperms, the average seed mass of angiosperm species is the largest near the 400 mm isohyet, which is also the boundary line between the wet zone and the arid zone of China. It is also the dividing line between the forest area and the grassland area, and the strong filtering effect of the habitat on the species in this area may lead to the existence of large seeds. Increases in seed production have been observed in forests at the steppe–forest boundary compared to mid-elevation forests with better relative soil moisture conditions [36].

Seed mass is a trait that exhibits significant phylogenic signals. This result has been repeatedly verified by previous studies, and the same results have been obtained from the community level to the global level, regardless of what scale the study is based on [6,37]. It is generally believed that during a period of rapid angiosperm diversification (85–65 Ma), angiosperms move out of the tropics and shift from being predominantly small-seeded to having a much wider range of seed mass [10,38]. Gymnosperms have a smaller seed mass

distribution range, but the seeds are generally larger than those of angiosperms, which is also consistent with our data distribution.

Life form has an important effect on seed mass. It is generally believed that multi-stem shrubs are mainly reproduced by sprouting, while single-stem trees are mainly reproduced by seeds [39,40]. Species that are reproduced by sprouting produce fewer and lower seed masses compared to seed reproduction [41]. Therefore, the seed mass of shrubs is significantly lower than that of trees, which may be due to the change in reproductive mode, and shrubs are more inclined to store energy for sprouting germination rather than for seed production. The seed mass of fleshy fruits is also significantly higher than that of other fruit types. Studies have shown that there is a significant positive correlation between seed mass and fruit mass [42,43], and the fruit mass of fleshy fruits is significantly higher than that of other fruit types. It has been suggested that the formation of large seeds and fleshy fruits co-evolved [8]. The cone seed mass produced by the gymnosperms is between that of the several fruits of the angiosperms.

Seed mass is also closely related to the mode of seed dispersal, with gravity-dispersed seeds being the largest and wind-borne seeds being the smallest. The seed mass, tree height, and the mode of dispersal jointly determine how far a seed can travel [41,44]. Due to their reliance on wind dispersal, species with a small seed mass tend to have greater terminal velocities [45]. At the same time, this also affects the relationship between wind speed and seed mass. The higher the wind speed, the lower the seed mass. It is not ruled out that trees in areas with a higher wind speed will rely more on wind dispersal.

### 4.2. Differences in Effects of Environmental Factors on Gymnosperm and Angiosperm Seed Mass

Mean annual temperature and annual precipitation are significantly strongly correlated with seed mass compared to other climatic metrics for both gymnosperm and angiosperm seed masses. Precipitation and temperature are two of the most commonly considered variables in studies of how the climate shapes plant community distribution and diversity on a global scale [46,47], and they are often used to simulate and predict the impact of climate change on forest vegetation [48]. It is generally considered that higher metabolic costs arise at higher temperatures, as metabolic expenditure (respiration for growth and maintenance) increases with temperature [49], thus also promoting the production of larger seeds [50,51]. Since the study site is chosen in China, the areas with a high temperature tend to have more precipitation, and the areas with low temperature also have less precipitation [52,53]. Therefore, temperature and precipitation cause consistent changes in seed mass.

Although previous studies have suggested that annual precipitation has a much smaller effect on seed mass [46,50], it still has a certain regulatory effect on plant traits [54]. A comparison of the results of field experiments and surveys with different precipitation gradients showed that short-term increases in precipitation promote the production of small seeds, while long-term increases in precipitation (a cross-regional study) favor the production of large seeds [54]. The community structure in different precipitation regions is different, and it is generally believed that as precipitation increases, the coverage of woody species increases, which may lead to light becoming an increasingly important limiting resource. Furthermore, large seeds are more likely to germinate in light and poor communities, which will help increase the competitiveness of large-seeded plants, so areas with high precipitation tend to produce larger seeds [7,10]. Compared with winter precipitation, the warmest season (generally from May to August in China) often coincides with the tree growth season and has a greater impact on tree growth and reproduction. Our results show an overall increasing trend in seed mass with increasing precipitation, as well as a significant increase in the upper values of gymnosperm and angiosperm seed masses. Our results are also consistent with previous works which indicate that after long-term adaptation, with the increase in precipitation, large seeds are still more competitive [55,56], which shows that climatic aridity filters species distribution through seed mass.

Annual precipitation is often used to assess plant responses to the environment, though it is not exactly equal to available water. In practical applications, the seasonal distribution of rainfall, evaporation, and other factors need to be considered, as well as the process of rainfall from canopy interception to soil moisture infiltration [57]. The aridity index was chosen to represent consistent effects of atmospheric water deficits in our study. Although the effect of the aridity index on seed mass was not as significant as that of annual mean temperature and annual precipitation, there were differences in the response of gymnosperm and angiosperm seed masses to drought. Compared with angiosperms, the correlation between seed mass and the drought index was stronger in gymnosperms, and the drought index alone explained more of seed mass. Our results indicate that the seed mass changes in gymnosperms are more sensitive than those of angiosperms to climatic aridity. Briefly, dry climate filters gymnosperms but not angiosperms through seed mass, and gymnosperm seed masses are concentrated in a certain range in arid regions. Studies have shown that there is a significant phylogenetic signal in plant drought tolerances [58], and some tree ring evidence also shows differences in the sensitivity of gymnosperms and angiosperms to drought [59–61]. They also reflect that gymnosperms and angiosperms may have different survival strategies in arid regions, and seed mass is also generally considered an important trait that reflects plant adaptation [62]. The study on multi-plant traits also found that compared with gymnosperms, angiosperms were more functionally diverse and displayed more diverse strategies, including high drought tolerance, high shade tolerance, and high growth rates [62], which also matches the results of our research on seed mass.

In summary, our study focused on differences in the interspecific seed masses of gymnosperm and angiosperm responses to changes in environmental factors, providing a reference for understanding the functional fitness of plants in a changing world. With the change in environmental factors, gymnosperm seed mass changes are more sensitive, indicating that environmental factors have a greater impact on gymnosperm seed mass than angiosperm.

**Supplementary Materials:** The following supporting information can be downloaded at: https://www.mdpi.com/article/10.3390/d15030401/s1, Figure S1: The phylogenetic tree generated using the 'V.PhyloMaker' package in R.; Table S1: Relationship between average, minimum and maximum seed mass and environmental factors.

**Author Contributions:** H.L. and Y.Q. conceived of the study idea. B.H. contributed some data to this article. Y.Q. performed the analyses. H.L., Y.Q., C.X. and J.D. interpreted the results and implications. H.L. supervised the research. Y.Q. and H.L. wrote the first draft of the manuscript. All authors revised the text and provided critical feedback. All authors have read and agreed to the published version of the manuscript.

**Funding:** This work was granted by National Key Research and Development Program of China (Grant No. 2022YFF0801803).

**Institutional Review Board Statement:** Not applicable.

**Data Availability Statement:** The data that support the findings of this study are available from the corresponding author upon reasonable request.

**Conflicts of Interest:** The authors declare no conflict of interest.

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
