# Peer review of "Dry Climate Filters Gymnosperms but Not Angiosperms through Seed Mass"

_diversity, doi:10.3390/d15030401_

Round 1
Reviewer 1 Report
This study have tried to analyze the phylogenetic signals with seed mass and climatic factors. It included mean annual temperature, annual precipitation, solar radiation, and potential evapotranspiration as climatic variables to examine the relationship between phylogenetic signals and seed mass. The results showed the seed mass has a significant phylogenetic signal, which is in consistent with recent studies. However, the seed mass was no significant difference between gymnosperms and angiosperms. The seed mass of angiosperms has a positive correlation with all of the climatic factors; the seed mass of gymnosperms only significantly correlated to the annual mean temperature and annual precipitation. However, some studies have suggested that annual precipitation may have limited effects on seed mass. As an alternative, I suggest the authors may consider using bioclimatic variables that provide a better indication of seed production during the growing season, such as the precipitation in the warmest quarter (bio18) or the wettest quarter (bio16), as these may have more ecological significance. However, recent studies also mentioned that the functional traits, such as growth forms , dispersal mode, leaf area, etc., are better predictors of seed mass than climatic variables (e.g.:; Zheng et al. 2017 Scientific Reports 7: 2741).
In the drier climate, the seed mass of gymnosperms tended to become light and with low variability, and more sensitive than angiosperms to climatic aridity. Yet the aridity index can be a measure to assess the water balance in a given climatic region. However, it should be noted that there can be significant variability in precipitation levels and that droughts may not occur immediately, but rather have a time lag. Alternatively, standardized precipitation evapotranspiration index (SPEI) may be a better candidate to show the long-term trends. In addition, SPEI is standardized and make it easier to compare drought conditions across different regions. I suggest if it is possible, SPEI may be used for a predictor.
In general, the result sounds, but the manuscript needs to be improved before accept. The followings are the point-to-point questions and suggestions:
-
P1 abstract: I suggest revising the abstract to include a short description of the analysis procedures used in your study, which would be helpful to the readers.
-
P1 L25–27 “The different patterns of seed mass along a climate aridity gradient between gymnosperms and angiosperms might point to different future fates for these two taxonomic groups, which has not yet been considered in dynamic vegetation modeling.” I think aridity gradient is an important factors to the seed mass of angiosperms/gymnosperms, but the “future fates” is an ambiguous concept. I cannot easily figured out what will happen for gymnosperms from the above conclusion. Please try to address concrete and concise statements of your results, such as the reproduction rate of gymnosperm would decrease under a drier climate. However, the connection between seed mass and dynamic vegetation modeling is also weak and ambiguous.
-
P2 L92: There are different versions of WorldClim dataset. I suggest to add the version number, such as WorldClim version 2.
-
P2 L94–95: Data citation is crucial, especially for large dataset producers. While you have provided a URL from CGIAR-CSI, it is important to also include a proper citation for the water balance dataset: Zomer, R.J.; Xu, J.; Trabuco, A. (2022) Version 3 of the Global Aridity Index and Potential Evapotranspiration Database. Scientific Data 9, 409. https://www.nature.com/articles/s41597-022-01493-1. However, the data source description is insufficient. The CGIAR-CSI stands for “Consultative Group on International Agricultural Research-Consortium for Spatial Information” (note: the latest version is 6, please also add the version number)
-
P3 L103: Please add citations of Blomberg’s K and Pagel’s \lambda.
-
p3 L107–108: What is the definition of “spatial structure of the data”? Do you mean to reduce the influences of “spatial autocorrelation”? Why test the significance of your regression result using t-test can reduce the “spatial structure of the data”? How to perform it? It suppose to calculate the Moran’s I to examine the spatial autocorrelation (if it is what you mean) and if involving the spatial data, the general t-test and simple linear regression can not be used because the spatial autocorrelation violate the statistical consumption.
-
p3 L125: Why the n=1580? The total number of individuals is 2,575. Did you have any preprocess of the data?
-
p6 L179–190: The mean annual temperature and annual precipitation were found to be strongly correlated with seed mass in this study, and similar results have been reported in other studies. It is thought that higher temperatures may promote the production of larger seeds. The study site in China tends to have more precipitation during periods of high temperature and vice versa (L188–189). And then? I expect to read more about your discussion based on your discovery, but it suddenly ended here. Did you have further discussion about this?
-
p6 L188: “Since the study site is chosen in China, the areas with high temperature tend to have more precipitation, and the areas with low temperature also have less precipitation”: Do you have citation?
-
Please check and confirm the format requirements of references, such as
-
The inline citation should use square brackets, such as [1–2]
-
The order of references should be based on the order in which the cited literature appear
-
The journal names should be use abbreviated ones
-
Book type should add pages (ex: L283–284)
-
L355: The author “Station SNFF” should use the full name
-
p279: larix sibirica —> Larix sibirica; kazakhstan —> Kazakhstan
-
Author Response
Please see the attachment for the revised main text. The following are the replies to the review comments point-by-point.
- As an alternative, I suggest the authors may consider using bioclimatic variables that provide a better indication of seed production during the growing season, such as the precipitation in the warmest quarter (bio18) or the wettest quarter (bio16), as these may have more ecological significance. However, recent studies also mentioned that the functional traits, such as growth forms , dispersal mode, leaf area, etc., are better predictors of seed mass than climatic variables (e.g.:; Zheng et al. 2017 Scientific Reports 7: 2741).
Reply: Thank you very much for your suggestions. We have added research on the effect of functional traits (growth type, fruit type, dispersal mode) on seed mass in section 3.3 of the article results, and added the precipitation in the warmest quarter (bio18) as an environmental factor affecting seed mass.
- However, it should be noted that there can be significant variability in precipitation levels and that droughts may not occur immediately, but rather have a time lag. Alternatively, standardized precipitation evapotranspiration index (SPEI) may be a better candidate to show the long-term trends. In addition, SPEI is standardized and make it easier to compare drought conditions across different regions. I suggest if it is possible, SPEI may be used for a predictor.
Reply: Thank you very much for your suggestion. Since we did not have a specific sampling year for seed mass, more precise climate indicators were not used. In addition, SPEI and others mainly consider climate events, we analyze them according to the spatial climate gradient.
- P1 abstract: I suggest revising the abstract to include a short description of the analysis procedures used in your study, which would be helpful to the readers.
Reply: Thanks for the suggestion, the analysis method has been supplemented in the abstract
- P1 L25–27 “The different patterns of seed mass along a climate aridity gradient between gymnosperms and angiosperms might point to different future fates for these two taxonomic groups, which has not yet been considered in dynamic vegetation modeling.” I think aridity gradient is an important factors to the seed mass of angiosperms/gymnosperms, but the “future fates” is an ambiguous concept. I cannot easily figured out what will happen for gymnosperms from the above conclusion. Please try to address concrete and concise statements of your results, such as the reproduction rate of gymnosperm would decrease under a drier climate. However, the connection between seed mass and dynamic vegetation modeling is also weak and ambiguous.
Reply: Thank you for your suggestion, the original sentence has been revised.
- P2 L92: There are different versions of WorldClim dataset. I suggest to add the version number, such as WorldClim version 2.
Reply: Already edited.
- P2 L94–95: Data citation is crucial, especially for large dataset producers. While you have provided a URL from CGIAR-CSI, it is important to also include a proper citation for the water balance dataset: Zomer, R.J.; Xu, J.; Trabuco, A. (2022) Version 3 of the Global Aridity Index and Potential Evapotranspiration Database. Scientific Data 9, 409. https://www.nature.com/articles/s41597-022-01493-1. However, the data source description is insufficient. The CGIAR-CSI stands for “Consultative Group on International Agricultural Research-Consortium for Spatial Information” (note: the latest version is 6, please also add the version number)
Reply: Thank you for your suggestion, it has been modified.
- P3 L103: Please add citations of Blomberg’s K and Pagel’s \lambda.
Reply: Already edited.
- p3 L107–108: What is the definition of “spatial structure of the data”? Do you mean to reduce the influences of “spatial autocorrelation”? Why test the significance of your regression result using t-test can reduce the “spatial structure of the data”? How to perform it? It suppose to calculate the Moran’s I to examine the spatial autocorrelation (if it is what you mean) and if involving the spatial data, the general t-test and simple linear regression can not be used because the spatial autocorrelation violate the statistical consumption.
Reply: We changed the data processing method and averaged the seed mass into the grid, which can reduce certain errors when considering the distribution of species.
- p3 L125: Why the n=1580? The total number of individuals is 2,575. Did you have any preprocess of the data?
Reply: Thanks for the suggestion, relevant information and treatments have been added in M&M.
- p6 L179–190: The mean annual temperature and annual precipitation were found to be strongly correlated with seed mass in this study, and similar results have been reported in other studies. It is thought that higher temperatures may promote the production of larger seeds. The study site in China tends to have more precipitation during periods of high temperature and vice versa (L188–189). And then? I expect to read more about your discussion based on your discovery, but it suddenly ended here. Did you have further discussion about this?
Reply: The Discussion section is supplemented with ”Since the study site is chosen in China, the areas with high temperature tend to have more precipitation, and the areas with low temperature also have less precipitation [55,56]. Therefore temperature and precipitation caused consistent results in the change of seed mass.”
- p6 L188: “Since the study site is chosen in China, the areas with high temperature tend to have more precipitation, and the areas with low temperature also have less precipitation”: Do you have citation?
Reply: References have been added.
- Please check and confirm the format requirements of references, such as
The inline citation should use square brackets, such as [1–2]
The order of references should be based on the order in which the cited literature appear
The journal names should be use abbreviated ones
Book type should add pages (ex: L283–284)
L355: The author “Station SNFF” should use the full name
p279: larix sibirica —> Larix sibirica; kazakhstan —> Kazakhstan
Reply: References have been normalized.

Reviewer 2 Report
The authors presented an interesting perspective to investigate the relationship between environmental factors and plant functional traits with a dataset of tree seed mass in China, from which phylogenetic signal was detected in seed mass and an identifiable difference in the response to aridity between the evolutionarily distinctive angiosperms and gymnosperms were shown. This is an important and much needed addition to the evidence base for our comprehensive understanding of the diversity and functions in ecosystems around the globe. However, due to potential statistical flaws (details in later comments), shall the authors’ original analysis design remain unchanged, care should be taken when discussing this result in the context of environmental filtering for the two plant groups and future climate change.
Specific comments:
(1) In addition to reporting the total number of individuals (i.e. tree specimens) from the data, to present a better picture of the scope of this study, the authors should include a map of their locations and a list of species involved this sample of 2575 individuals, at least they should report a total number of species for angiosperms and gymnosperms. For best detail level, I suggest the authors submit a supplementary table where each row reports a tree species with their original binomials and number of individuals in their data, plus the accepted scientific name, genus and family checked against TPL.
(2) Fig. S2 should be in the main text to present a full context of this study, as it showed more than “no significant difference between seed mass of gymnosperms and angiosperms in overall comparison (line 121)”, there might be a potential difference in seed mass distribution. The mass distribution of gymnosperm seeds seems to be positively skewed (with more seeds to the lighter end), just judging intuitively from diagram itself. Given the seed mass was log-transformed (please report the base of the logarithm at line 98, I assume it was 10 to make best sense), the actual difference in mass could be vast. The authors should report the statistical test in the Methods section of this “significance”, and consider using other tests that enable comparing the variance, skewness, or any reasonable stats other than the mean/median of the two groups if possible. Additionally, despite citing evidences on significant differences between seed mass in angiosperms and gymnosperms (line 42), the authors did not call back to this point and interpret this result in the Discussion, which can be more thoroughly explored together with the phylogenetic signal detected in this study.
(3) Simple linear regressions generally performed poorly in teasing apart confounding and potentially competing environmental variables in this study, as no R-squared value exceeded 0.2 in the results regressing log-seed mass against MAT, AP, SR and AI in any plant groups. Consequently, a comparison in the size of R-squared was not compelling to identify a true independent environmental effect, not even enough given hierarchical partitioning was performed with only the four predictor variables involved.
First, the authors should report the spatial resolution of the data acquired from worldclim.org, and the collinearity or pairwise correlation between the environmental variables should be taken into the design of their analysis, and I do not see how a t-test of OLS parameters can “reduce the influence of spatial structure” (lines 107-109) without introducing some spatial-autocorrelation measure. Second, since phylogenetic signal was present in the data, the authors should also consider teasing apart the intra- and inter-taxa variance of seed mass, which might also be related to environmental filtering or acclimation in different spatial scales. Last, a more comprehensive model explaining the seed mass in the data should be built taking into account of the following aspects: independent groups of the environmental predictors and their spatial distribution, plus the inherent difference among different taxa.
If the observed variance in seed mass were not properly partitioned in the model, result of regressions, the presented OLS and quantile regressions included, could be merely a statistical artefact (e.g. there were possibly only a few under-sampled gymnosperm taxa in the arid regions, and the angiosperms have larger variance in all seed traits across any environmental gradient in general because of the sheer size of the sample) rather than an important insight. A clear difference in gymnosperms and angiosperms in terms of their seed mass response to drought and relevant discussions of the two plant groups’ evolution and acclimation is still promising and attractive, but please consult with a statistician to improve this current analysis before saying that.
(4) Words from Onstein (2020) were cited improperly at line 60, because angiosperms do not produce cones like gymnosperms do, please consider rephrasing this sentence.
(5) The authors should be more consistent referring to their “aridity index” constructed as MAP/PET (lines 93-96), hence greater magnitude in this value indicated more hydric environments. It was called “humidity” at line 149 and shortly after described as “decrease of aridity index” at line 151. In the Discussion, terms like “drought” (line 213) and “climatic aridity” (line 219) were used instead. Please consider reducing the number of interchangeable terms used in this paper and rephrase the Results section in more plain words to avoid confusion.

Author Response
Please see the attachment for the revised main text. The following are the replies to the review comments point-by-point.
- In addition to reporting the total number of individuals (i.e. tree specimens) from the data, to present a better picture of the scope of this study, the authors should include a map of their locations and a list of species involved this sample of 2575 individuals, at least they should report a total number of species for angiosperms and gymnosperms. For best detail level, I suggest the authors submit a supplementary table where each row reports a tree species with their original binomials and number of individuals in their data, plus the accepted scientific name, genus and family checked against TPL.
Reply: Thanks for the suggestion, relevant information and treatments have been added in M&M.
- S2 should be in the main text to present a full context of this study, as it showed more than “no significant difference between seed mass of gymnosperms and angiosperms in overall comparison (line 121)”, there might be a potential difference in seed mass distribution. The mass distribution of gymnosperm seeds seems to be positively skewed (with more seeds to the lighter end), just judging intuitively from diagram itself. Given the seed mass was log-transformed (please report the base of the logarithm at line 98, I assume it was 10 to make best sense), the actual difference in mass could be vast. The authors should report the statistical test in the Methods section of this “significance”, and consider using other tests that enable comparing the variance, skewness, or any reasonable stats other than the mean/median of the two groups if possible. Additionally, despite citing evidences on significant differences between seed mass in angiosperms and gymnosperms (line 42), the authors did not call back to this point and interpret this result in the Discussion, which can be more thoroughly explored together with the phylogenetic signal detected in this study.
Reply: We have added analyses of gymnosperm and angiosperm seed mass skewness, kurtosis, normality in section 3.2. At the same time, after considering the distribution of species, the error caused by uneven sampling of gymnosperms and angiosperms can be reduced to a certain extent
- Simple linear regressions generally performed poorly in teasing apart confounding and potentially competing environmental variables in this study, as no R-squared value exceeded 0.2 in the results regressing log-seed mass against MAT, AP, SR and AI in any plant groups. Consequently, a comparison in the size of R-squared was not compelling to identify a true independent environmental effect, not even enough given hierarchical partitioning was performed with only the four predictor variables involved.
Reply: I added research on the effect of functional traits (growth type, fruit type, dispersal mode) on seed mass in section 3.3 of the results, and added the precipitation in the warmest quarter (bio18) and wind speed as an environmental factor affecting seed mass. Under the new analysis method, the correlation coefficients between environmental factors and seed masshave been improved.
- First, the authors should report the spatial resolution of the data acquired from worldclim.org, and the collinearity or pairwise correlation between the environmental variables should be taken into the design of their analysis, and I do not see how a t-test of OLS parameters can “reduce the influence of spatial structure” (lines 107-109) without introducing some spatial-autocorrelation measure. Second, since phylogenetic signal was present in the data, the authors should also consider teasing apart the intra- and inter-taxa variance of seed mass, which might also be related to environmental filtering or acclimation in different spatial scales. Last, a more comprehensive model explaining the seed mass in the data should be built taking into account of the following aspects: independent groups of the environmental predictors and their spatial distribution, plus the inherent difference among different taxa.
Reply: Thanks for the suggestion, we have remapped the spatial pattern of seed mass distribution of major gymnosperm and angiosperm woody plants in China in section 3.1, which can solve this problem to some extent.
- If the observed variance in seed mass were not properly partitioned in the model, result of regressions, the presented OLS and quantile regressions included, could be merely a statistical artefact (e.g. there were possibly only a few under-sampled gymnosperm taxa in the arid regions, and the angiosperms have larger variance in all seed traits across any environmental gradient in general because of the sheer size of the sample) rather than an important insight. A clear difference in gymnosperms and angiosperms in terms of their seed mass response to drought and relevant discussions of the two plant groups’ evolution and acclimation is still promising and attractive, but please consult with a statistician to improve this current analysis before saying that.
Reply: Thank you very much for your suggestion. We have considered your suggestion to include the species distribution in the analysis and obtained results that corroborate the previous conclusions.
- Words from Onstein (2020) were cited improperly at line 60, because angiosperms do not produce cones like gymnosperms do, please consider rephrasing this sentence.
Reply: The original sentence has been modified.
- The authors should be more consistent referring to their “aridity index” constructed as MAP/PET (lines 93-96), hence greater magnitude in this value indicated more hydric environments. It was called “humidity” at line 149 and shortly after described as “decrease of aridity index” at line 151. In the Discussion, terms like “drought” (line 213) and “climatic aridity” (line 219) were used instead. Please consider reducing the number of interchangeable terms used in this paper and rephrase the Results section in more plain words to avoid confusion.
Reply: Words that caused confusion have been revised.

Round 2
Reviewer 2 Report
The authors in this second draft provided additional information that enables readers to better understand the scope of their data, which is a welcoming change that made the analysis appear more robust. However, careful revision is still needed, especially the English language in the new inputs is not as fluent as in the previous draft, thus, some errors, inconsistencies, and confusions should be avoided before the final publication of the manuscript.
Specific comments:
(1) According to the second and third paragraph from section 2.1 of “Materials and Methods”, readers were led to assume that no exact seed collection location in the book "Chinese Woody Plant Seeds" or the Shandong Forest and Grass Germplasm Resource Center was used in the following analysis on the relationship between seed mass and the environmental variables, because all the species from this 2575-individual collection were checked against their distribution range available on cvh.ac.cn and essentially every species was assigned a single constant mean seed mass value across its whole range, which was defined as all its recorded locations on CVH. Please correct me if I misunderstood the description and try to rephrase it. If I’m not misunderstood, this treatment is acceptable as the study was mainly focused on interspecific variations, but please avoid referring to patterns associated with intraspecific variations while discussing the results, such as talking about seed mass acclimation to precipitation change, and please acknowledge the limitation of this data due to the unaccounted large spatial variation of seed mass in species with a large distribution range.
(2) Section 3.1 of the results (including Figure 4 and Table 3) need reorganization in order to more clearly present the regression of seed mass (mean, max, min) against the environmental variables for Gymnosperms and Angiosperms in the grids. First, please report the regression results of the mean seed mass in Table 3 rather than in words under the caption of Figure 4 (lines 359-364). Second, plotting out the regression lines of Amax and Amin, or Gmax and Gmin on the same panels would help readers understand the wordy descriptions in lines 322-325, which also need rephrasing, and please be careful with “distribution range” or “distribution”, as geographic distribution of species was also an important concept in this study and referring to the “data range” of the seed mass here would be more appropriate to avoid confusion. Third, reporting simple linear regressions as “correlations” were not recommended, as it is mathematically understandable but conceptually different. Please be consistent and at least not use “correlation” on the caption of Table 3.
(3) Lines 186-187, it shall be more concise to describe a “natural logarithmic transformation” than using more word to introduce the base as e.
(4) The Jonckheere-Terpstra test, introduced at line 195, was not an appropriate test for the difference of seed mass between the Gymnosperms and Angiosperms, because it assumed an ordinal comparison of the median values, and a more appropriate non-parametric test could be the Mann-Whitney U test.
(5) Lines 225-230, it is redundant to include a justification of an R package choice in the methods. Moreover, it did not justify using hierarchical partitioning for the sake of this analysis aside from “more researchers are using it in ecology”, if the choice were the final decision, please just be frank and concise.
(6) The authors mentioned the “400mm isoprecipitation line” multiple times, in the Abstract and the main text (at lines 239 and 442), citing its significance as a boundary between the wet and arid zones in China. Please include some citations for this statement, and discuss a bit more on how this pattern could be interpreted in the observed relationship between angiosperm seed mass and precipitation, if any.
(7) Lines 255-257, the result description is ambiguous. Please choose simple sentences with consistent phrasing.
(8) Lines 289-290, avoid using “gravity-borne” when referring to dispersal strategy (also at line 492). Using animal or wind-borne is ok, but please be consistent with other parts of the whole manuscript and only use “gravity dispersed”, “animal dispersed”, etc.
(9) Lines 314 and 320, check typos.
(10) Line 431, ambiguous wording “was also higher”.
(11) Lines 452-453. This statement is not true. The results recorded “no significant difference” between the seed mass of gymnosperms and angiosperms despite a more appropriate test could be used. In addition, according to the distribution in Figure 2, gymnosperms actually had more seeds with mass lower than the median. Or, were the authors referring to the seed size here, which was not reported in the results of this study?
(12) Lines 539-540, and 583-584, the authors discussed “seed quality” multiple times, and from what I understood no other functional traits than seed mass were meant in those phrases. Could it be a mistranslation of “seed mass”?
